

# EFT matching from analyticity and unitarity

**Stefano De Angelis[1⋆] and Gauthier Durieux[2†]**

**1** Institut de Physique Théorique, CEA, CNRS, Université Paris-Saclay,
F-91191 Gif-sur-Yvette cedex, France
**2** Theoretical Physics Department, CERN, 1211 Geneva 23, Switzerland

⋆ stefano.de-angelis@ipht.fr , † gauthier.durieux@cern.ch

## Abstract

We present a new on-shell method for the matching of ultraviolet models featuring massive states onto their massless effective field theory. We employ a dispersion relation in the space of complex momentum dilations to capture, in a single variable, the relevant analytic structure of scattering amplitudes at any multiplicity. Multivariate complex analysis and crossing considerations are therefore avoided. Remarkably, no knowledge about the infrared effective field theory is required in dimensional regularisation. All matching information is extracted from the residues and iscontinuities of the ultraviolet scattering amplitudes, which unitarity expresses in terms of lower-point and lower-loop results, respectively. This decomposition into simpler building blocks could deliver new insight in the structure of the effective field theories obtained from classes of ultraviolet scenarios and facilitate computations at higher loop orders.

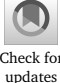

# 1   Introduction

Effective field theories (EFTs) are becoming the prime interpretation framework for collider data. The lack of unambiguous sign of new resonance at the energy frontier, together with the upcoming increases in luminosity and precision rather than energy, indeed motivates an indirect approach to hypothetical heavy new physics. Understanding how specific ultraviolet (UV) models —and classes thereof— populate the infrared (IR) EFT parameter space, and grasping the implications of EFT results for concrete scenarios is however essential. The procedure of matching between full models and their low-energy EFTs establishes the needed UV–IR correspondence.

The automation of matching computations has recently been pushed to the one-loop level [1–5] using diagrammatic [6, 7] and functional [8–32] methods, together with the method of regions in dimensional regularisation [33, 34]. Here, we devise a distinct approach which relies on a dispersion relation in the complex plane of momentum dilations. It expresses the low-energy EFT amplitudes in terms of the tree-propagator residues and loop cuts of UV amplitudes. Thereby, unitarity reduces the complexity of the amplitudes required for matching, in the number of legs or loops. By exploiting simpler building blocks, it may moreover deliver new insight.

The required computations are facilitated by the successful developments of unitarity methods [35–45] and Feynman diagram integration techniques (see *e.g.* [46, 47] for detailed reviews and references therein) in dimensional regularisation [48] applied to high-multiplicity and high-order calculations in perturbation theory, for gauge theories [49, 50] and gravity [51, 52]. Scattering-amplitude methods are also finding more and more applications in the study of EFTs and, in particular, that of the Standard Model (SMEFT). They helped uncovering positivity constraints [53–55], non-interferences between renormalisable and dimension-six amplitudes [56], and non-renormalisation theorems between higher-dimensional operators [57–61]. They provided an alternative means of computing anomalous dimensions [62–70], of constructing SMEFT operator bases [71–77] as well as broken-phase massive amplitudes [78–86].

Our on-shell approach to matching was inspired by the computation of Wilson coefficients directly from unitarity cuts in [87] and from the procedure used to derive positivity constraints on operator coefficients in terms of dispersion relations [88]. These hinted at the possibility of expressing the Wilson coefficients of an IR EFT in terms of the discontinuities of UV amplitudes. However, in the context of positivity bounds, only four-point amplitudes are accessible and the analytic structure in a single Mandelstam invariant is most often exploited. We use form factors [62] to control analytic properties at arbitrary multiplicity (they were first studied on-shell in the context of the $\mathcal{N} = 4$ super Yang-Mills theory [89, 90]). Moreover, we simplify the multivariate complex analysis into a one-variable problem using an analytically continued momentum dilation parameter, which is reminiscent of the BCFW shift [91].

Before providing a more rigorous discussion in section 3, we start by presenting the basic principles of this new matching method in section 2. A working example of scalar theory is discussed in section 4, and conclusions are presented in section 5.

# 2   Basic principles

For simplicity, let us focus here on an IR EFT containing only massless scalars. Matching would make the IR and UV amplitudes identical for small enough external momenta $p_i$:

$$\mathcal{A}_{\text{IR}} = \mathcal{A}_{\text{UV}}, \quad \text{for} \quad p_i/M \to 0, \tag{1}$$

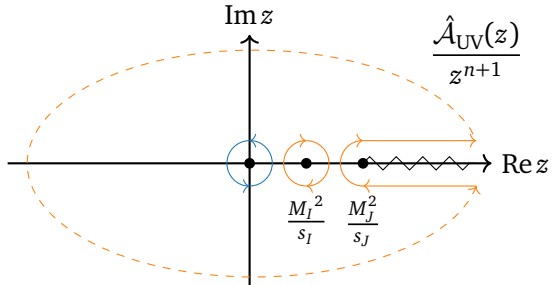

Figure 1: Each $n$th order of the tree-level IR EFT amplitude expanded in powers of Mandelstam invariants is expressed as a contour integral of the subtracted UV amplitude $\hat{\mathcal{A}}_{\mathrm{UV}}(z)/z^{n+1}$ in the complex plane of momentum dilations. All matching information is therefore extracted from the residues and branch cuts of the UV amplitude alone. For illustration, we picture one residue at $z = M_I^2/s_I$ arising from the $1/(s_I - M_I^2)$ pole of a tree propagator and one branch cut starting at $z = M_J^2/s_J$ which could for instance arise from a $\log(M_J^2 - s_J)$ term.

where $M$ is the mass scale of the heavy UV states that are integrated out. As matching condition, we enforce the term-by-term equality of the amplitudes expanded at all orders in powers of external momenta (squared into Mandelstam invariants). On the IR EFT side, this expansion involves the tree-level contact terms which readily map onto operators and are multiplied by the Wilson coefficients to be matched.[1] From a purely on-shell perspective, these tree-level amplitudes of higher-and-higher multiplicities fully characterise the EFT and allow to reconstruct it entirely.

To count powers of squared external momenta, let us introduce momentum dilations, acting on Mandelstam invariants and amplitudes as

$$s_I \to \hat{s}_I = z\,s_I, \quad \text{and} \quad \mathcal{A} \to \hat{\mathcal{A}}(z), \tag{2}$$

where $I$ is a unordered subset of external particles and $s_I \equiv (\sum_{i \in I} p_i^\mu)^2$. Analytically continuing $z$ to the complex plane, the power-by-power matching is enforced by equating the $z = 0$ residues of the two amplitudes divided by an integer powers of $z$:

$$\operatorname*{Res}_{z=0} \frac{\hat{\mathcal{A}}_{\mathrm{IR}}(z)}{z^{n+1}} = \operatorname*{Res}_{z=0} \frac{\hat{\mathcal{A}}_{\mathrm{UV}}(z)}{z^{n+1}}. \tag{3}$$

Note that negative $n$'s are allowed but that the associated residues vanish unless $\hat{\mathcal{A}}(z)$ itself contains poles at $z = 0$, corresponding to massless factorisation channels.

On the IR EFT side, the residues straightforwardly extract terms of the tree-level amplitude homogenous in the Mandelstam invariants, *e.g.* $c_n s_I^n + c_n' s_I^{n+1}/s_J$. On the UV side, let us express each residue as an integral over a small contour surrounding the origin, at $z = 0$ (in blue on Figure 1). Deforming this contour towards infinity extracts the residues and discontinuities of $\mathcal{A}_{\mathrm{UV}}$ in all Mandelstam invariants (in orange on Figure 1). Indeed, the non-analycities in $z$ are inherited from the poles and branch cuts in Mandelstam invariants: $s_I = M_I^2$ poles give rise to $z = M_I^2/s_I$ ones, and $s_J \geq M_J^2$ branch cuts give rise to $z \geq M_J^2/s_J$ ones. The contour at infinity may also yield a non-vanishing contribution if $n$ is small enough.

Unitarity can then be used to express these residues and discontinuities in terms of lower-point and lower-loop amplitudes. The matching at tree level only involves pole residues which

---

[1]As will be seen more rigorously below, our power-by-power matching does not capture contributions which do not admit a Laurent expansion at zero external momenta. Heuristically, dimensionally regulated loops in our scaleless EFT have branch points in this limit, *i.e.* no zero-momentum expansion, and do not contribute. On the IR EFT side, the Laurent expansion therefore only captures tree-level contributions.

can be expressed as products of two lower-point on-shell amplitudes:

$$\mathcal{A}_{\text{IR}}^{\text{tree}} \supset -\sum_{\text{poles}} \operatorname*{Res}_{z=M_I^2/s_I} \frac{\hat{\mathcal{A}}_{\text{UV}}(z)}{z^{n+1}} = \sum_{z=M_I^2/s_I} \frac{\hat{\mathcal{A}}_{\text{UV}}^{\text{left}}(z)\hat{\mathcal{A}}_{\text{UV}}^{\text{right}}(z)}{s_I \, z^{n+1}} \,. \tag{4}$$

At the one-loop level, unitarity turns discontinuities into two-particle cuts involving two tree-level amplitudes and an intermediate phase-space integration:

$$\begin{aligned}
\mathcal{A}_{\text{IR}}^{\text{tree}} &\supset +\frac{1}{2\pi i} \sum_{\substack{\text{branch}\\\text{cuts}}} \int_{\frac{M_J^2}{s_J}}^{\infty} \frac{dz}{z^{n+1}} \left(\hat{\mathcal{A}}_{\text{UV}}(z+i\epsilon) - \hat{\mathcal{A}}_{\text{UV}}(z-i\epsilon)\right) \\
&= \frac{1}{2\pi} \sum_{\substack{\text{branch}\\\text{cuts}}} \int_{\frac{M_J^2}{s_J}}^{\infty} \frac{dz}{z^{n+1}} \int \text{dLIPS} \, \hat{\mathcal{A}}_{\text{UV}}^{\text{left}}(z)\hat{\mathcal{A}}_{\text{UV}}^{\text{right}}(z) \,,
\end{aligned} \tag{5}$$

with the appropriate symmetry factor left implicit on the right-hand side.

Subtleties omitted in the above discussion and addressed below are the following:

- A form factor with an additional momentum influx $q$ has to be considered instead of the UV amplitude to confine all non-analyticities to known locations, namely on the positive real $z$ axis.

  Note the polynomial dependence of the obtained result in Mandelstam invariants renders the crossing and $q \to 0$ limit trivial. Calculations at four points and above can therefore in practice be performed directly on amplitudes.

- The UV amplitude may have branch cuts extending all the way down to $z = 0$, in which case one needs to formally consider the expansion around a small $z = -\delta < 0$. After taking $\delta \to 0$, the IR divergence of the cut integral can be handled, as customary, within dimensional regularisation.

**Simplest examples** Before closing this section, let us give simple examples for which the subtleties above are irrelevant (more complicated cases are discussed in section 4). Let us consider a $\phi^3\Phi$ theory, involving a massless scalar $\phi$ and a heavy $\Phi$ of mass $M$ to be integrated out. The tree-level exchanges

$$\begin{gathered}
\overset{i}{\underset{j}{\diagdown}}\overset{g_4 \quad g_4}{\underset{k}{\diagup}} \\
\end{gathered} \tag{6}$$

generate six-point contact terms in the IR EFT, which can trivially be obtained from the lower-point $\mathcal{A}(\phi\phi\phi\Phi) = g_4$ amplitude:

$$\mathcal{A}_{\text{IR},6}^{\text{tree},(0)} \supset \sum_{z=M^2/s_{ijk}} \frac{\hat{\mathcal{A}}(\phi\phi\phi\Phi)^2}{s_{ijk} \, z^{n+1}} = \frac{g_4^2}{M^2} \sum_{10 \, (ijk) \, \text{perm.}} \left(\frac{s_{ijk}}{M^2}\right)^n \,, \tag{7}$$

where the sum on the right-hand side runs over the 10 independent unordered permutations of $ijk$ external legs. At the loop level, four-point contact terms are generated by the bubbles

$$\begin{gathered}
\overset{i}{\underset{j}{\diagdown}}g_4 \, \bigcirc\bigcirc \, g_4 \diagup \\
\end{gathered} \tag{8}$$

whose cut can be expressed as the square of the tree-level $\mathcal{A}(\phi\phi\phi\Phi)$ amplitude and a two-particle $\phi\Phi$ Lorentz invariant phase-space integral, $\int \text{dLIPS} = \frac{1}{8\pi}(1 - \frac{M^2}{z s_{ij}})$. The last ingredient

is the $z$ integral along the cut:

$$
\mathcal{A}_{\text{IR},4}^{\text{tree},(1)} \supset \frac{1}{2\pi} \sum_{s_{ij}=s,t,u} \int_{M^2/s_{ij}}^{\infty} \frac{\mathrm{d}z}{z^{n+1}} \int \mathrm{dLIPS}\, \hat{\mathcal{A}}(\phi\phi\phi\Phi)^2
$$
$$
= \frac{g_4^2}{16\pi^2 n(n+1)} \sum_{s_{ij}=s,t,u} \left(\frac{s_{ij}}{M^2}\right)^n, \quad \text{for} \quad n>0. \tag{9}
$$

The renormalisable $n=0$ term is UV divergent and requires a regulator to be computed (see section 4).

Remarkably, both at tree and loop levels, the EFT amplitude is obtained at once, to all orders in the derivative expansion, thereby fixing a whole tower of Wilson coefficients.

## 3   Matching formula

We aim to match a UV theory involving heavy states to the corresponding EFT of light IR states only. For simplicity, we focus on massless light states.[2] After matching, the EFT truncated at a given order should approximate the predictions of the UV theory for momenta much smaller than the heavy masses.

The central objects of our matching procedure are the form factors of a local operator $\mathcal{O}$, having a momentum influx $q$, a set of $m$ particles carrying quantum numbers $\vec{m}$ as out-state, and the vacuum as in-state,[3]

$$
F_{\mathcal{O}}(\vec{m}) = \int \mathrm{d}^d x\, e^{ix\cdot q}\, {}_{\text{out}}\langle\psi_{\vec{m}}|\mathcal{O}(x)|0\rangle = (2\pi)^d\, \delta^{(d)}(q-p_1\cdots-p_m)\, {}_{\text{out}}\langle\psi_{\vec{m}}|\mathcal{O}(0)|0\rangle, \tag{10}
$$

which are formally defined as the limiting value of a complex function $F_{\mathcal{O}}(s_{ij}+i\epsilon)$ using the Feynman-$i\epsilon$ prescription. In particular, we consider form factors of the Lagrangian operator which are the closest to (purely on-shell) S-matrix elements. Matching to all orders in inverse powers of the heavy mass scale $M$ would result in identical UV and IR predictions

$$
F_{\mathcal{L}_{\text{IR}}}(\vec{m}) = F_{\mathcal{L}_{\text{UV}}}(\vec{m}), \tag{11}
$$

for $q^2 = (p_1+...+p_m)^2 < M^2$, where $\mathcal{L}_{\text{UV}}$ includes the modes that are dynamical at high energies, while $\mathcal{L}_{\text{IR}}$ only contains light modes interacting through higher-dimensional local operators.

As sketched in section 2, we realise this matching by imposing an order-by-order identification in the momentum expansion or, equivalently, in powers of a momentum dilation variable $z$. Let us thus consider the continuous form-factor deformation $F_{\mathcal{O}}(\vec{m};z)$, by the complex momentum dilation (2), away from the physical $z = 1+i\epsilon$ configuration. Since all external particles are outgoing, the Mandelstam invariants are all positive, $s_I \geq 0$. This ensures that all the form-factor singularities on the physical sheet of the complex $z$ plane are confined to the

---

[2]In the presence of massive external states, ensuring momentum conservation and on-shell conditions would naively require masses to be dilated in addition to momenta. Non-analyticities in $z$ would therefore be introduced which are not related to non-analyticities in Mandelstam invariants (e.g. arising from logarithms of the masses). It remains to be examined how those could be consistently handled.

[3]It is worth emphasising that, when we consider form factors, there is no relation between the different kinematic invariants arising from momentum conservation: $(p_1+...+p_n)^2 = q^2 \neq 0$. In particular, it is necessary to consider $q^\mu$ time-like. Alternatively, form factors can be thought of as the decay amplitude of a very massive (non-dynamical) scalar of square mass equal to $q^2$. In practice, we will however consider $q \neq 0$ only in the three-point case. At and above four points, we can set $q=0$ after performing crossing of some of the out-states into the in-state, with the proper normalisation. The form factors are strictly related to the S-matrix elements by the simple relation ${}_{\text{out}}\langle\psi_{\vec{n}}|\mathcal{O}(0)|\psi_{\vec{m}}\rangle_{\text{in}}|_{q^2=0} = \partial_g\, {}_{\text{out}}\langle\psi_{\vec{n}}|\psi_{\vec{m}}\rangle_{\text{in}}$ where $g$ is the coupling of the operator $\mathcal{O}$ in the Lagrangian.

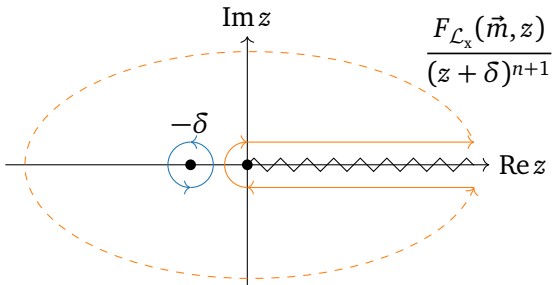

Figure 2: Analytic structure of $F_{\mathcal{L}_{\text{x}}}(\vec{m}, z)/(z+\delta)^{n+1}$ in the complex $z$ plane of momentum dilations. We isolate terms of the Taylor expansion in $z + \delta$ by integrating on a small contour around $z = -\delta$, before taking $\delta \to 0$. This small contour is then deformed towards infinity, thereby capturing the discontinuities and residues of the form factor on the positive real axis.

positive real axis (see *e.g.* [62]). The discontinuity across the $z \geq 0$ branch cut is then given by the sum of the physical discontinuities in each Mandelstam invariant:

$$\text{Disc}_{z} F_{\mathcal{O}}(\vec{m}; z) = \sum_{I} \text{Disc}_{s_I} F_{\mathcal{O}}(\vec{m}; z), \tag{12}$$

where $I$ is an unordered subset of external particles. The discontinuity across a $m_I$-particle cut is given by the product of form-factors and amplitudes (at lower loop order, in perturbation theory), integrated over the appropriate Lorentz-invariant phase space [92]:

$$\text{Disc}_{s_I} F_{\mathcal{O}}(\vec{m}; 1 + i\epsilon) = i \sum_{X} \frac{(-1)^{m_X}}{S_X} \int \text{dLIPS}_X \, F_{\mathcal{O}}(\vec{m}_{\bar{I}}, \vec{m}_X; 1 - i\epsilon) \mathcal{A}(\vec{m}_X \to \vec{m}_I), \tag{13}$$

where the sum runs over all possible sets of internal states of $m_X$ particles in the cut and where the external state is partitioned as $\vec{m} = \{\vec{m}_I, \vec{m}_{\bar{I}}\}$. Similarly, the residues are directly related to the physical residues appearing in Mandelstam invariants:

$$\text{Res}_{z = M_I^2/s_I} F_{\mathcal{O}}(\vec{m}; z) = \frac{1}{s_I} \text{Res}_{s_I = M_I^2} F_{\mathcal{O}}(\vec{m}; 1) \bigg|_{s_J \to s_J M_I^2/s_I}, \tag{14}$$

which are given by the factorisation onto lower-point form factors and amplitudes:

$$\text{Res}_{s_I = M_I^2} F_{\mathcal{O}}(\vec{m}; 1 + i\epsilon) = -\sum_{X} F_{\mathcal{O}}(\vec{m}_{\bar{I}}, X; 1 - i\epsilon) \mathcal{A}(X \to \vec{m}_I), \tag{15}$$

where the sum is over all possible one-particle state exchanges.

In $z$-space, the identification between the UV and IR theories can be imposed within the smallest $|z| < M_I^2/s_I$ radius. To avoid possible singularities at $z = 0$, we formally perform our power-by-power matching of the form factors expanded around $z = -\delta$ for a small $\delta > 0$, where analyticity is guaranteed. Our matching condition therefore becomes:

$$\mathcal{P}_n^{\text{IR}}(\vec{m}) = \mathcal{P}_n^{\text{UV}}(\vec{m}),$$

$$\text{with} \qquad \mathcal{P}_n^{\text{x}}(\vec{m}) \equiv \lim_{\delta \to 0} \text{Res}_{z = -\delta} \frac{F_{\mathcal{L}_{\text{x}}}(\vec{m}, z)}{(z+\delta)^{n+1}} = \lim_{\delta \to 0} \frac{1}{2\pi i} \oint_{-\delta} \frac{\text{d}z}{(z+\delta)^{n+1}} F_{\mathcal{L}_{\text{x}}}(\vec{m}, z). \tag{16}$$

The contour integral can then be deformed towards $|z| \to \infty$ as in Figure 2, which captures the non-trivial analytic structure of $F_{\mathcal{L}_{\text{IR}}}(\vec{m}; z)$ and $F_{\mathcal{L}_{\text{UV}}}(\vec{m}; z)$ along the positive real axis.

On the IR side of this matching condition (16), we obtain

$$
\begin{aligned}
\mathcal{P}_n^{\text{IR}}(\vec{m}) = &+ \frac{1}{2\pi i} \int_0^\infty \frac{dz}{(z+\delta)^{n+1}} \operatorname*{Disc}_z F_{\mathcal{L}_{\text{IR}}}(\vec{m};z) \\
&- \frac{1}{\delta^{n+1}} \operatorname*{Res}_{z=0} F_{\mathcal{L}_{\text{IR}}}(\vec{m};z) \\
&- \operatorname*{Res}_{z=\infty} \frac{F_{\mathcal{L}_{\text{IR}}}(\vec{m};z)}{z^{n+1}} ,
\end{aligned}
\tag{17}
$$

where the $\delta \to 0$ limit is understood. It is particularly advantageous to regulate all divergences using dimensional regularisation. Contributions from the discontinuities then become scaleless integrals in $z$ and vanish. The loop contributions to the residue at infinity similarly vanish: The residue at infinity is tree-level exact. Indeed, as the IR theory has no mass scale, the dilated form factor is a sum of terms each homogenous in $z$. In $d = 4-2\epsilon$ dimensions, the integrands in both cases have a $z^{\alpha-\epsilon}$ form for some integer $\alpha$ and therefore vanish for a suitable choice of dimensional regularisation parameter $\epsilon$. A more detailed discussion is provided in Appendix B. Therefore, the residue at infinity of the full form factor is that of the tree-level one, which can in turn be identified with the one at $z=0$ since there is no other non-analyticity at tree level:

$$
-\operatorname*{Res}_{z=\infty} \frac{F_{\mathcal{L}_{\text{IR}}}(\vec{m};z)}{z^{n+1}} = -\operatorname*{Res}_{z=\infty} \frac{F_{\mathcal{L}_{\text{IR}}}^{\text{tree}}(\vec{m};z)}{z^{n+1}} = +\operatorname*{Res}_{z=0} \frac{F_{\mathcal{L}_{\text{IR}}}^{\text{tree}}(\vec{m};z)}{z^{n+1}} .
\tag{18}
$$

So (17) simplifies to

$$
\mathcal{P}_n^{\text{IR}}(\vec{m}) = -\frac{1}{\delta^{n+1}} \operatorname*{Res}_{z=0} F_{\mathcal{L}_{\text{IR}}}(\vec{m};z) + \operatorname*{Res}_{z=0} \frac{F_{\mathcal{L}_{\text{IR}}}^{\text{tree}}(\vec{m};z)}{z^{n+1}} .
\tag{19}
$$

Let us now consider the UV side of the matching condition (16). Again, we express the residue at $z = -\delta$ as a small contour integral subsequently deformed towards $|z| \to \infty$ as in Figure 2, which captures the non-analyticities of $F_{\mathcal{L}_{\text{UV}}}(\vec{m};z)$:

$$
\begin{aligned}
\mathcal{P}_n^{\text{UV}}(\vec{m}) = &- \sum_I \left(\frac{s_I}{M_I^2}\right)^{n+1} \operatorname*{Res}_{z=M_I^2/s_I} F_{\mathcal{L}_{\text{UV}}}(\vec{m};z) \\
&+ \frac{1}{2\pi i} \int_0^\infty \frac{dz}{z^{n+1}} \operatorname*{Disc}_z F_{\mathcal{L}_{\text{UV}}}(\vec{m};z) \\
&- \frac{1}{\delta^{n+1}} \operatorname*{Res}_{z=0} F_{\mathcal{L}_{\text{UV}}}(\vec{m};z) \\
&- \operatorname*{Res}_{z=\infty} \frac{F_{\mathcal{L}_{\text{UV}}}(\vec{m};z)}{z^{n+1}} ,
\end{aligned}
\tag{20}
$$

where the $\delta \to 0$ limit is still understood. Possible non-analyticities are poles and branch points at $z = M_I^2/s_I$ generated by massive tree propagator and loop cuts, a massless tree propagator pole and loop branch point at $z=0$, and a pole at infinity. The $\delta \to 0$ limit of the $z \geq 0$ discontinuity integral is by definition singular, but the associated singularities can be traded for singularities in the IR regulator as $\delta$ is taken to zero. More details are provided in Appendix B. The residue at $z=0$ is the only term in which a $\delta$ dependence remains. As it is generated by massless IR poles, it however cancels against the analogous term in the IR part of the matching in (19). The residue at infinity is again tree-level exact, as the $z$ integral is again scaleless in the $|z| \to \infty$ limit (see Appendix B).

Imposing the matching condition $\mathcal{P}_n^{\text{IR}}(\vec{m}) = \mathcal{P}_n^{\text{UV}}(\vec{m})$ and cancelling the $1/\delta^{n+1}$ terms between the UV and the IR, one can therefore extract each term of the tree-level expansion of the IR form factor from the non-analyticities of the UV form factor:

$$\operatorname*{Res}_{z=0} \frac{F_{\mathcal{L}_{\mathrm{IR}}}^{\mathrm{tree}}(\vec{m};z)}{z^{n+1}} = -\sum_{I}\left(\frac{s_I}{M_I^2}\right)^{n+1} \operatorname*{Res}_{z=M_I^2/s_I} F_{\mathcal{L}_{\mathrm{UV}}}(\vec{m};z)$$

$$+ \frac{1}{2\pi i}\int_0^\infty \frac{\mathrm{d}z}{z^{n+1}} \operatorname*{Disc}_z F_{\mathcal{L}_{\mathrm{UV}}}(\vec{m};z) \tag{21}$$

$$- \operatorname*{Res}_{z=\infty} \frac{F_{\mathcal{L}_{\mathrm{UV}}}^{\mathrm{tree}}(\vec{m};z)}{z^{n+1}}.$$

We emphasise that, combining the analytic properties of the form factors and dimensional regularisation, we managed to isolate the tree-level IR form factor, bypassing the subtraction of IR loops (which contribute to the renormalisation-group running). Note that our master formula (21) is also valid for $n$ negative, in which case the factorisable components of the tree-level form factor (having poles at $z = 0$) are also extracted from the UV. In practice, both sides of (21) however vanish identically, providing no matching information, unless

$$n \ge p, \quad \text{for} \quad p \equiv \frac{\min\left\{d - m\left(\frac{d}{2}-1\right) - [c_{\mathrm{IR}}]\right\}}{2}, \tag{22}$$

where $d$ is the space-time dimension and $[c_{\mathrm{IR}}]$ is the total mass dimension of the couplings appearing in the IR form factor. The full tree-level IR form factor can thus be reconstructed from the sum of (21) over $n$:

$$F_{\mathcal{L}_{\mathrm{IR}}}^{\mathrm{tree}}(\vec{m}) = \sum_{n=p}^{\infty} \operatorname*{Res}_{z=0} \frac{F_{\mathcal{L}_{\mathrm{IR}}}^{\mathrm{tree}}(\vec{m};z)}{z^{n+1}}. \tag{23}$$

The computation of a residue at infinity is scarcely needed. Dimensional analysis implies that it vanishes unless $n \le \max(d - m(d/2-1) - [c_{\mathrm{UV}}])/2$. Combined with the condition (22), it is thus only necessary when $\min[c_{\mathrm{UV}}] \le d - m(d/2-1) - 2n \le \max[c_{\mathrm{IR}}]$. The most relevant local operator coefficient leading to four- and higher-points contact-term amplitudes in the IR is marginal ($\max[c_{\mathrm{IR}}] \le 0$), just as the most irrelevant coupling of a renormalisable UV theory ($0 \le \min[c_{\mathrm{UV}}]$). In these particularly relevant cases, the computation of a residue at infinity is therefore at most required for a single value of $n$ saturating the inequalities above, and for the extraction of a marginal coupling. Since only the leading high-energy component of the renormalisable UV amplitude is then needed, this computation can moreover be performed in a strict massless limit (assuming that it is well-defined).[4] Instead, if the UV theory is non-renormalisable, $\min[c_{\mathrm{UV}}]$ can be negative and the computation of a residue at infinity can be needed for more than one value of $n$. Masses then also become relevant.

The formula (21) above constitutes the main result of this paper. It refines the argument of [87] that one-loop rational terms are not relevant for the matching at four points. Indeed, the $z$ discontinuity of these terms only arises in non-integer dimensions, from a $z^\epsilon$ dependence, and vanish in the $\epsilon \to 0$ limit. On the other hand, for higher-point loop amplitudes, rational terms may give additional contributions to the pole residues of $F_{\mathcal{L}_{\mathrm{UV}}}$ and thus contribute to the matching.

Remarkably, this matching procedure, combined with a purely on-shell construction of the full IR amplitude, does not require the specification of an EFT operator basis. Unless Wilson coefficients defined from a Lagrangian are desired, no information about the IR is actually needed and all matching information is extracted from the UV alone. The full UV amplitude

---

[4]Note that the contributions from the arc at infinity comes purely from the UV, contrary to the case of positivity bounds from $2 \to 2$ scattering amplitudes where one can have high-energy scattering and small scattering angle (large distance). Indeed, we are in the physical kinematic configuration for which $q^2 = s + t + u > 0$ and $s \sim t \sim u$.

is not even required since only lower-point factorisations and lower-loop cuts are necessary. If present, the residue at infinity only needs to be evaluated at tree level and in the high-energy limit. Matching is thus expressed in terms of simpler building block which may also allow to uncover new patterns and selection rules in the EFT of UV models.

Finally, as a by-product of our analysis, we have also generalised the method of [62] for computing anomalous dimensions from S-matrix elements with internal massive states (see Appendix A). The decoupling of heavy modes in the renormalisable group evolution is manifest within this framework.

## 4 Scalar working example

In this section, we study the matching of a toy scalar UV model using our central formula (21). This scalar theory exhibits all the singularity structures (coming from Feynman integrals) potentially appearing in our matching procedure. Additional subtleties arising from spinors (*e.g.* evanescent operators) and tensor structures (*e.g.* gauge and gravitational theories) will be discussed elsewhere.

This toy scalar theory features a single heavy scalar $\Phi$ of mass $M$ to be integrated out, and a massless $\phi$ which remains dynamical in the low-energy EFT. It is defined by the following Lagrangian:

$$\mathcal{L}_{\text{UV}} = \frac{1}{2} \partial_\mu \phi \partial^\mu \phi + \frac{1}{2} \partial_\mu \Phi \partial^\mu \Phi - \frac{1}{2} M^2 \Phi^2 - \frac{\lambda}{4!} \phi^4 - \frac{g_3}{2!} \Phi \phi^2 - \frac{g_4}{3!} \Phi \phi^3 \,. \tag{24}$$

Radiative corrections will generate a $\phi^3$ term (for example, at order $g_3 g_4$) because the $g_4$ interaction breaks the $\mathbb{Z}_2$ symmetry of $\phi$. In the following, we however focus on higher-point interactions for which we can take the momentum influx of form factors to zero ($q \to 0$) and effectively consider amplitudes in different channels, after crossing.

**Tree-level four points**   Let us start with the four-$\phi$ tree-level amplitude:

$$
\begin{array}{cc}
\phantom{x} & \phantom{x}
\end{array}
\tag{25}
$$

Although we do not need to write them down explicitly when proceeding on-shell, let us give the UV amplitude and the expected form of the IR one:

$$\mathcal{A}_{\text{UV},4}^{\text{tree}} = \lambda - \sum_{s,t,u} \frac{g_3^2}{s_{ij} - M^2} \,, \tag{26}$$

$$\mathcal{A}_{\text{IR},4}^{\text{tree}} = \lambda + \sum_{n=0}^{\infty} g_3^2 \frac{c_n}{M^{2n+2}} (s^n + t^n + u^n) \,, \tag{27}$$

where $c_n$ are the Wilson coefficients to be determined.

In this tree-level example, only the residues appearing in the master formula (21) need to be evaluated. The massive propagator gives rise to poles at $z = M^2/s_{ij}$ (for $s_{ij} = s, t, u$) whose residues are just the product of two $\mathcal{A}(\phi\phi\Phi) = g_3$ amplitudes:

$$\mathcal{A}_{\text{IR},4}^{\text{tree},(0)} \supset - \sum_{s_{ij}=s,t,u} \underset{z=\frac{M^2}{s_{ij}}}{\text{Res}} \frac{\hat{\mathcal{A}}_{\text{UV},4}^{\text{tree}}(z)}{z^{n+1}} = \sum_{z=\frac{M^2}{s,t,u}} \frac{\mathcal{A}(\phi\phi\Phi)^2}{M^2 z^n} = \frac{g_3^2}{M^{2n+2}} (s^n + t^n + u^n) \,, \tag{28}$$

which implies that $c_n = 1$. Note that in a scalar theory, $-p$ defined in (22) equivalently counts the number of massless propagators. Since no massless propagator is present in this case (*i.e.* $p = 0$), one only needs to consider $n \geq 0$.

For completeness, note that the residue at infinity yields the extra quartic coupling dependence of the $n = 0$ term of $\mathcal{A}_{\text{IR},4}^{\text{tree},(0)}$ expansion:

$$\mathcal{A}_{\text{IR},4}^{\text{tree},(0)} \supset -\operatorname*{Res}_{z=\infty} \frac{\hat{\mathcal{A}}_{\text{UV},4}^{\text{tree}}(z)}{z} = \lambda. \tag{29}$$

**Tree-level six points**   Let us now go to six points, still at the tree level. UV amplitude topologies (the last one was already examined in section 2) are:

$$\tag{30}$$

where we ignore a $g_3^4$ contribution, for simplicity. The second and third topologies generate residues at $z = M^2/s_{ij}$ and $M^2/s_{ijk}$ which are picked up in our matching formula, giving:

$$\mathcal{A}_{\text{IR},6}^{\text{tree},(0)} \supset \sum_{20\ (ijk)\ \text{perm.}} -\frac{1}{s_{ijk}} \frac{\lambda g_3^2}{M^2} \frac{s_{ij}^{n+1} + s_{jk}^{n+1} + s_{ki}^{n+1}}{M^{2n+2}}, \qquad \text{and} \qquad \sum_{20\ (ijk)\ \text{perm.}} \frac{1}{2} \frac{g_4^2}{M^2} \left(\frac{s_{ijk}}{M^2}\right)^n. \tag{31}$$

In the first case, only $n \geq \min(-2 - 2[\times])/2 = -1$ is needed, while in the second case $n \geq \min(-2 - [\times])/2 = 0$ is sufficient.

The first topology in (30) only generates a renormalisable contribution which appears as a residue at infinity for $n = \min(-2 - 2[\times])/2 = -1$:

$$\mathcal{A}_{\text{IR},6}^{\text{tree},(0)} \supset \sum_{20\ (ijk)\ \text{perm.}} -\frac{1}{2} \frac{\lambda^2}{s_{ijk}}. \tag{32}$$

As in the second part of (31), a symmetry factor of $1/2$ is included since only half of the $(ijk)$ permutations are independent.

**One-loop four points**   At the one-loop level, the four-point topologies are the following:

$$\tag{33}$$

ignoring again terms of order $g_3^4$, for simplicity.

The first one was considered already in section 2. Let us re-examine it in $d$ dimensions, to be able to also extract its $n = 0$ renormalisable component which diverges in four dimensions. In the first line of (9), we thus need to evaluate the $d$-dimensional phase-space integral:

$$\int \mathrm{dLIPS}_d = \int \frac{\mathrm{d}^d l}{(2\pi)^{d-2}} \delta^+(l^2 - M^2) \delta^+((p_i + p_j - l)^2)$$
$$= \frac{1}{(16\pi)^{1-\epsilon}} \frac{s_{ij}^{-\epsilon}}{\Gamma[\frac{3}{2} - \epsilon]} \left(1 - \frac{M^2}{s_{ij}}\right)^{1-2\epsilon}, \tag{34}$$

with $\delta^+(l^2 - m^2) \equiv \theta(l^0) \delta(l^2 - m^2)$. After dilation and integration over the branch cut, one then gets:

$$\int_{\frac{M^2}{s_{ij}}}^{\infty} \frac{\mathrm{d}z}{z^{n+1}} \int \mathrm{dLIPS}_d = \left(\frac{s_{ij}}{M^2}\right)^n \frac{\sqrt{\pi}\,\Gamma[2 - 2\epsilon]\Gamma[n + \epsilon]}{(16\pi)^{1-\epsilon}\,\Gamma[\frac{3}{2} - \epsilon]\Gamma[n + 2 - \epsilon]} M^{-2\epsilon}, \tag{35}$$

which leads to the result obtained in section 2:

$$\mathcal{A}_{\text{IR},4}^{\text{tree},(1)} \supset \frac{g_3^2}{16\pi^2 n(n+1)} \left(\frac{s_{ij}}{M^2}\right)^n, \tag{36}$$

in the four-dimensional limit and for $n > 0$. For $n = 0$, one gets

$$\mathcal{A}_{\text{IR},4}^{\text{tree},(1)} \supset \frac{g_3^2}{16\pi^2} \left(\frac{1}{\bar{\epsilon}} + \log\frac{\mu^2}{M^2} + 1\right), \qquad \text{where} \qquad \frac{1}{\bar{\epsilon}} \equiv \frac{1}{\epsilon} - \gamma_E + \log 4\pi. \tag{37}$$

The UV divergence can be cancelled by a quartic coupling counterterm, and it is easy to check that the results above match those obtained from the hard-region expansion of the loop integral.

The second and third topologies in (33) give rise to branch cuts extending down to $z = 0$. The second topology also generates a residue at $z = M^2/s_{ij}$, proportional to a bubble loop function. Together with the integral along the discontinuity of the bubble and triangle loops, we thus obtain:

$$\begin{aligned}
\mathcal{A}_{\text{IR},4}^{\text{tree},(1)} \supset \; & + \lambda g_3^2 \frac{s_{ij}^n}{M^{2n+2}} B(M^2 - i\epsilon; 0, 0) \\
& + \frac{\lambda g_3^2}{2\pi} \int_0^\infty \frac{dz}{z^{n+1}} \frac{-1}{zs_{ij} - M^2 + i\epsilon} \int d\text{LIPS}_d \\
& + \frac{\lambda g_3^2}{2\pi} \int_0^\infty \frac{dz}{z^{n+1}} \int d\text{LIPS}_d \left(-\frac{1}{(l - \sqrt{z}p_i)^2 - M^2} - \frac{1}{(l - \sqrt{z}p_j)^2 - M^2}\right),
\end{aligned} \tag{38}$$

where a factor of $2 \times 1/2$ arising from the exchange of initial and final states and from the Bose symmetry of the light scalars in the loop is understood. The massless bubble integral $B(P^2; 0, 0)$ is:

$$B(P^2; 0, 0) = -\frac{\pi^{\frac{d}{2}} (-P^2)^{\frac{d}{2}-2}}{(4\pi)^{d-\frac{3}{2}} \sin\frac{\pi d}{2} \, \Gamma\left[\frac{d-1}{2}\right]}, \tag{39}$$

and the $M^2 - i\epsilon$ prescription is fixed by the position of the $z = M^2/s - i\epsilon$ pole. The on-shell integration measure can be parametrised as

$$d^d l \, \delta^+(l^2) = dl^0 \, \theta(l^0) \, \delta((l^0)^2 - l^2) \, dl \, l^{d-2} d\cos\theta \, (\sin\theta)^{d-4} \frac{2\pi^{d/2-1}}{\Gamma[\frac{d}{2} - 1]}, \tag{40}$$

with $l^\mu = (l^0, l\,\vec{\Omega}_{d-2}, l\cos\theta)$ and note that the two contributions in the last line of (38) are identical. One can explicitly check that the first two lines of (38) add up to zero, *i.e.* the residue and the discontinuity of the second diagram in (33) cancel each other. This should not be a surprise, as the hard-region expansion of the bubble diagram from which they both originate corresponds to a scaleless integral, as discussed in Appendix B. To perform the cut-triangle integral, we follow the integration strategy outlined in [93] and find

$$\int_0^\infty \frac{dz}{z^{n+1}} \int d\text{LIPS}_d \frac{1}{(l - p_i)^2 - M^2} = \left(\frac{s_{ij}}{M^2}\right)^n \frac{M^{d-6}}{8(4\pi)^{\frac{d}{2}-2}} \frac{(-1)^{n+1} n! \csc\frac{\pi d}{2}}{\Gamma[\frac{d}{2} + n]}, \tag{41}$$

which agrees with the hard-region expansion of the full integral. Then, (38) becomes

$$\mathcal{A}_{\text{IR},4}^{\text{tree},(1)} \supset \frac{\lambda g_3^2}{16\pi^2 M^2} \frac{(-1)^n}{n+1} \left(\frac{s_{ij}}{M^2}\right)^n \left(\frac{1}{\bar{\epsilon}} + H_{n+1} + \log\frac{\mu^2}{M^2} + \mathcal{O}(\epsilon)\right), \tag{42}$$

where $H_n \equiv \sum_{k=1}^{n} \frac{1}{k}$ are the harmonic numbers. As discussed in Appendix B, the soft-regions of the bubble and triangle loops ($l^2 \sim s_{ij} \ll M^2$), which encode the renormalisation-group running of the EFT, do not contribute to our matching formula.

The result (42) exhibits a $1/\epsilon$ pole at any $n$, which is understood as an IR divergence associated with the mass of the heavy state in the UV theory. The associated logarithm does not contribute to the running of the EFT, as shown in Appendix A. Such divergences however always match UV divergences of opposite sign in the EFT (see, for example, the reviews [94, 95]). In the present case, this is clear because the full triangle integral is both UV and IR finite.

Note that the results obtained from the computation of cuts in a single channel may not be polynomial in Mandelstam invariants as those of (36), (37) and (42). In general, loops may have discontinuities in multiple channels (*e.g.* the box appearing in the four-point matching at order $g_3^4$) giving rise to transcendental functions of ratios of Mandelstam invariants. Only the sum over all channels is guaranteed to be polynomial and reproduces the hard-region expansion of the loop integrals.

## 5 Conclusions

We presented a new method for the matching of UV models onto their massless EFT. Relying only on on-shell quantities, it avoids the gauge and field-redefinition redundancies arising in the Lagrangian formalism. A dispersion relation in the space of complex momentum dilations is applied to form factors and captures the relevant analytic structure at any multiplicity and for generic kinematics. The IR tree-level amplitudes, which fully characterise the EFT and can be employed to bootstrap it, are extracted from the residues and discontinuities of the UV amplitudes regularised dimensionally. Contributions to all EFT orders are extracted at once. No operator basis or EFT computation is required, unless Wilson coefficients defined from a Lagrangian are desired. The full UV amplitudes are not even needed since unitary expresses their non-analyticities in terms of lower-point and lower-loop results. The possible contribution from a residue at infinity is tree-level exact, mostly needed to extract renormalisable contributions, and can be computed in the strict massless limit of a renormalisable UV theory.

The main computational difficulty arises from the evaluation of phase-space integrals across loop cuts, especially when the loop involves several uncut propagators (cut bubbles are trivial) and masses. Understanding how to introduce the method of regions at the level of such integrals, drawing inspiration from recent analyses [96–99], could ease such calculations. On the other hand, our conclusion that no other information than non-analyticities is needed for EFT matching could also be used to facilitate the computation at higher orders with traditional techniques.

By expressing matching in terms of simpler lower-point and lower-loop building blocks, our procedure may shed light on the structure of EFTs obtained from classes of UV models and make manifest selection rules. Examples of matched Wilson coefficients exhibiting *magic zeros*, with no immediate symmetry explanation, have notably been discussed in the literature recently [87, 100, 101]. Finally, since our method enables the dispersive extraction of Wilson coefficients appearing in amplitudes of any multiplicity, it may open the door to the derivation of positivity constraints in scatterings beyond the 2-to-2 case.

## Acknowledgements

We would like to thank Samuel Abreu, Miguel Correia, Giulia Isabella, Diego Redigolo, and especially Brando Bellazzini and David Kosower for stimulating discussions, as well as Mikael Chala, Christophe Grojean, Yael Shadmi and Chia-Hsien Shen for comments on our draft.

**Funding information** SDA acknowledges the hospitality of the CERN Theoretical Physics Department, where this work was initiated. SDA's research is supported by the European Research Council, under grant ERC-AdG-88541.

## A Renormalisation-group running with intermediate massive states

When considering a generic $z$, we know that

$$F_{\mathcal{O}}(\vec{m}; z) = z^{\frac{D}{2}} F_{\mathcal{O}}(\vec{m}; 1 + i\epsilon), \tag{A.1}$$

where $D = \sum_i p_i^\mu \frac{\partial}{\partial p_i^\mu}$ is the dilation operator. Homogeneity in mass dimension[5] tells us that

$$D = d_{F_{\mathcal{O}}} - \sum_i [g_i] g_i \frac{\partial}{\partial g_i} + D_\mu, \tag{A.2}$$

where $d_{F_{\mathcal{O}}} = \dim \mathcal{O} - m$ is the mass dimension of the $m$-point form factor, $\dim \mathcal{O}$ is the classical dimension of the operator considered, $g_i$ are the couplings of the theory, $[g_i]$ their dimension and

$$D_\mu = -\mu \frac{\partial}{\partial \mu}, \tag{A.3}$$

is also usually referred to as the *(anomalous) dilation operator*, as it only differs from $D$ by classical dimensions. The renormalisation scale $\mu$ is introduced in dimensional regularisation and controls the UV anomalous dimension of the operator (and the IR anomalous dimensions of the external states) as well as the beta functions of the couplings, through the Callan-Symanzik equation [102, 103].

In particular, combining equation (A.1) with unitarity, one finds

$$F_{\mathcal{O}}(\vec{m}; 1 + i\epsilon) = e^{-i\pi D} F_{\mathcal{O}}(\vec{m}; 1 - i\epsilon) = \sum_n {}_{\text{out}}\langle \psi_{\vec{m}} | \psi_{\vec{n}} \rangle_{\text{in}} F_{\mathcal{O}}(\vec{n}; 1 - i\epsilon). \tag{A.4}$$

In the second equality, the optical theorem was used by introducing a complete set of on-shell states $\psi_{\vec{n}}$ (and an integral over the associated phase space is left implicit). This is the central

---

[5]The dimension of the asymptotic states, defined as free-theory states, is fixed and does not run. If the asymptotic states are not well-defined because of long-range interactions, anomalous IR divergences appear in the Callan-Symanzik equation.

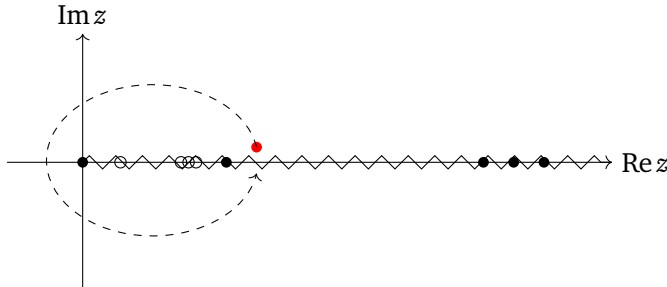

Figure 3: Analytical structure in $z$ of a three-point form factor, with massive particles in the spectrum of the theory. From left to right, the empty circles correspond to $z = M^2/s_{123}$ and the three $M^2/s_{ij}$. The filled one are located at $z = 4M^2/s_{123}$ and $4M^2/s_{ij}$. All the $s_{ij} = 0$ and $s_{123} = 0$ singularities are superimposed at $z = 0$.



formula proven in [62], which has found various applications in the computation of SMEFT anomalous dimensions [60,63,64,66] and in the formulation of non-renormalisation theorems at the two-loop level and beyond [58].

Considering a theory of massive particles and performing a $2\pi$ rotation of the form factor in $z$-space (see Figure 3), we notice that the right-hand side of (A.4) now depends on the kinematic configuration, *i.e.* on which branch points arise to the left of $z = 1$ on the real axis. In other words, for a fixed external kinematic configuration, only the discontinuities from the cuts with branch points at $z < 1$ contribute to the anomalous dimension, after modifying equation (A.2) to include contributions from the mass:

$$D = d_{F_{\mathcal{O}}} - M \frac{\partial}{\partial M} - \sum_i [g_i] g_i \frac{\partial}{\partial g_i} + D_\mu . \tag{A.5}$$

We can distinguish two cases:

- If the mass is smaller than the typical energies in the process, *i.e.* $M^2 < q^2$, there are a finite number of thresholds contributing to the running of the theory, up to $n$ intermediate massive states such that $n^2 M^2 > q^2$. Beyond that, the loops give virtual corrections which can be encapsulated into contact interactions. Desirable configurations to consider are either $n^2 M^2 > q^2$ or $n^2 M^2 < s_I$ for all the Mandelstam invariants, as otherwise the renormalisation and matching would be process-dependent and give non-local results. This is a strong version of the requirement to not have any hierarchy among the external momenta, when performing matching.

- Conversely, if we have $M^2 > q^2$, the heavy states decouples for every observables [104] and the thresholds do not contribute to the renormalisation group evolution of the theory [105]. This is equivalent to the ($\overline{\text{DS}}$) renormalisation scheme [106]. In this case, the hard regions of the loop integrals are analytic and the discontinuity vanishes. Then, only the soft region of integration contributes and the derivative expansion in $\partial^2/M^2$ determines the mixing of the EFT operators.

## B Soft loops and arcs at infinity as scaleless integrals

In this appendix, we explain how the loop discontinuities in the EFT and in the soft region of the UV theory, as well as the loop corrections to the arcs at infinity, are scaleless integrals within dimensional regularisation and therefore vanish. Explicit examples of such cancellations are seen in section 4.

We start from the first term in (17), which can be evaluated for generic values of $\delta$ in terms of an Euler integral:

$$
\begin{aligned}
\mathcal{I}_\delta &= \frac{1}{2\pi i} \int_0^\infty \frac{dz}{(z+\delta)^{n+1}} \operatorname*{Disc}_z F_{\mathcal{L}_{\text{IR}}}^\alpha (\vec{m}; z) \\
&= \sum_I \operatorname*{Disc}_{s_I} F_{\mathcal{L}_{\text{IR}}}^\alpha (\vec{m}) \bigg|_{s_J \to -s_J} \int_0^\infty \frac{dz}{(z+\delta)^{n+1}} (-z)^{\alpha-\epsilon} \\
&= \sum_I \operatorname*{Disc}_{s_I} F_{\mathcal{L}_{\text{IR}}}^\alpha (\vec{m}) \bigg|_{s_J \to -s_J} \frac{\Gamma[\alpha-\epsilon+1]\Gamma[\alpha+n+\epsilon]}{\Gamma[n+1]} \delta^{\alpha-n-\epsilon} ,
\end{aligned}
\tag{B.1}
$$

where $F_{\mathcal{L}_{\text{IR}}}(\vec{m}; z) = \sum_\alpha c_\alpha F_{\mathcal{L}_{\text{IR}}}^\alpha (\vec{m}; z)$, $c_\alpha$ are collections of couplings with fixed mass dimension and $\alpha$ is the corresponding mass dimension of the kinematic part of the form factor. In particular, we notice that the integral has a branch point at $\delta = 0$, as expected from the pinching of the contour of integration by the pole of the integrand at $z = -\delta$ and the branch point at

$z = 0$. A suitable choice of $\epsilon$ makes the $\delta \to 0$ limit finite and vanishing. We can thus take the defining values of $\lim_{\delta \to 0} I_\delta = 0$ and analytically continue to $\epsilon \sim 0$ in the complex plane to circumvent the poles generated by the $\Gamma$-functions on the real axis. An alternative approach is to split the integral on the second line of (B.1) into two pieces: from 0 to $\Lambda$ and from $\Lambda$ to $+\infty$. A suitable choice of the regulator (different for each integral) makes the two terms finite and equal up to an overall sign. Analytically continuing in $\epsilon$, their sum does not depend on $\Lambda$ and is vanishing.

A similar fate is shared by the soft-region contributions of the UV integrals with a branch cut starting at $z = 0$. This is clear from dimensional analysis, once we analyse the integrals using the method of regions. In the soft region, the loop momenta are of the order of the external momenta and much smaller than the heavy mass

$$l^2 \sim s_I \ll M^2 \,. \tag{B.2}$$

We can thus perform a Taylor expansion of the loop integrand in inverse powers of the heavy mass. After integration over loop momenta, each term of the expansion involves (dynamical) transcendental functions which depend of the ratio of Mandelstam invariants (which are all equally rescaled by $z$) and a kinematic factor which carries the transcendental dimensionality of the dimensionally regularised integral. The latter include all the $z$ dependence and is proportional to $z^{k-\epsilon}$ (where $k \in \mathbb{N}$). The $z$ integral along the branch cut is thus scaleless, and hence vanishing. We emphasise that such contributions, before integrating over $z$, determine the anomalous dimensions.

This does not mean that the loops with branch cuts starting at $z = 0$ in the UV theory do not contribute to the matching because the heavy mass introduces a new scale in the Feynman integrals. In the hard region characterised by

$$s_I \ll l^2 \sim M^2 \,, \tag{B.3}$$

the loop integrand can be expanded in powers of the external momenta. The transcendental dimension of the amplitude in this region is carried by the heavy-mass $M^{-2\epsilon}$ and the $z$ integral is not scaleless.

Then, equation (19) is strictly correct under two assumptions: There is no non-dynamical mass scale in the IR theory[6] (*i.e.* all the states are massless) and dimensional regularisation is employed. Relaxing any of these two assumptions restores the contributions from the EFT loops:

$$\mathcal{P}_n^{\text{IR}}(\vec{m}) \supset \frac{1}{2\pi i} \int_0^\infty \frac{\mathrm{d}z}{z^{n+1}} \, \underset{z}{\text{Disc}} \, F_{\mathcal{L}_{\text{IR}}}(\vec{m}; z) \,, \tag{B.4}$$

which would then cancel against additional contributions in the UV.

Similarly, we can show that the loop contribution from the arc at infinity vanishes in dimensional regularisation (see *e.g.* [107]). The form factor is a sum of terms each homogeneous in $z$ in this $|z| \to \infty$ limit. This is true even before the $|z| \to \infty$ limit for the massless EFT, while for the UV theory, it occurs after expanding in powers of $M_I^2/z s_I$. Then, the loop contributions regulated dimensionally give rise to integrals of the form:

$$\underset{z=\infty}{\text{Res}} \frac{F_{\mathcal{L}}^{\text{loop},\alpha}(\vec{m}; z)}{z^{n+1}} = \lim_{s_{ij}/M_{ij}^2 \to \infty} F_{\mathcal{L}}^{\text{loop},\alpha}(\vec{m}) \Big|_{s_J \to -s_J} \frac{1}{2\pi i} \oint_\infty \frac{\mathrm{d}z}{(-z)^{n+1-\alpha+\epsilon}}$$
$$\propto \lim_{r \to \infty} \frac{1}{r^{\alpha+n+\epsilon}} \,, \tag{B.5}$$

where $\epsilon$ must be chosen carefully to make such integral convergent, which is then vanishing.

---

[6]The integrals are not scaleless with dimensionful regularisation parameters or for EFTs with light massive states. Moreover, in the latter case we should be aware of the possibility of having anomalous thresholds in the $z$ complex plane. We leave this analysis for future investigation and limit ourselves to the study of massless EFTs.

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
