# Peer review of "EFT matching from analyticity and unitarity"

_SciPost Physics, doi:SciPost Phys. 16, 071 (2024)_

## Round 1 · Referee Report · Anonymous · 2023-11-2

Report

The article proposes a new approach to EFT matching for massless scalar theories for a given UV theory, based on dispersion relations. The matching takes place in the $z$ plane of dilatations, which rescales equally all Mandelstam variables. Subtractions in the EFT (where all heavy states are integrated out) allow to target Wilson coefficients, which are matched order by order to the same residues in the full theory. By deforming the contour and exploiting the knowledge of the analytic structure, those residues can be related to discontinuities and residues at infinity, which can be computed in the UV theory. The advantages of this method are: all matching takes place on-shell, avoiding all usual redundancies of working with a Lagrangian, furthermore the Wilson coefficients are extracted directly from discontinuities, hence knowledge of the full amplitude in the UV is not necessary. Finally, and in my opinion the most impressive achievement of this method, the choice of working in $z$ space allows to target $n$-point functions, as opposed to usual dispersion relations (such as those employed in positivity constraints), which are limited to $2\rightarrow 2$ scattering. The article comes with a precise derivation of the main formula and a few examples both at tree-level and one-loop.

In my opinion the idea is novel and nice and meets the standards for publication in SciPost. Despite this I find the paper extremely dense, technical and of difficult read, especially if the goal is to reach other groups that use more standard approaches to EFT matching and are not familiar with dispersion relations and overall analytical properties of the S-matrix (or form-factors). I leave to the authors the choice in what measure to address this aspect in the new version, but I think the following clarifications will make this work more approachable to its target audience, and should be addressed before publication.

1. The derivation of the matching formula is done for n-pt form factors, while all concrete examples of matching are on scattering amplitudes. Why is that? Also, a few lines describing and explaining the analytic structure of form factors in $z$ space would be useful for the readers, as it is not as commonly used as standard dispersion relations in $s$. Is this analytical structure unchanged in $D$ dimension, which is relevant as the contours are evaluated in dimensional regularization?

2. The fact that all particles in the EFT are massless (combined with dimensional analysis) is heavily used to simplify the dispersive form of $\mathcal{F}_{\mathcal{L}_{IR}}$ after eq. (17). Could you comment on what would happen if small masses would still be present and the integrals would not be scaleless? I would naively expect that the IR branch cut would cancel between the IR and the UV, and the expressions to be unchanged. Is this expectation too naive?

3. In the derivation of chapter 3 many subtle limits are taken. My understanding is that first the arc is deformed to infinity at $z\rightarrow \infty$, then the pole $\delta\rightarrow 0$ and lastly the limit of $D\rightarrow 4$. It would be useful to have a few sentences on this aspect: do these limits commute? The fact that some contours at infinity vanish in dimensional regularization would still be true if first the $D\rightarrow 4$ limit was taken?

4. Being able to extract Wilson coefficients from cuts is certainly nice, but an additional phase-space integral (and an integral over $z$) is still required. From the manuscript is not clear to me if you expect this method to be more computationally powerful than solving directly the loop integration (which has been explored and developed much more). It would be useful to clarify to the reader why this method would be advantageous in practical calculations.

  • validity: -
  • significance: -
  • originality: -
  • clarity: -
  • formatting: -
  • grammar: -

Author:  Stefano De Angelis  on 2023-12-07  [id 4177]

(in reply to Report 1 on 2023-11-02)

We would like to thank the referee for reading carefully our draft and for the interesting comments, which we address in the following.

The derivation of the matching formula is done for n-pt form factors, while all concrete examples of matching are on scattering amplitudes. Why is that? Also, a few lines describing and explaining the analytic structure of form factors in z space would be useful for the readers, as it is not as commonly used as standard dispersion relations in s. Is this analytical structure unchanged in D dimension, which is relevant as the contours are evaluated in dimensional regularization?

Since results are polynomial in Mandelstam invariant, the crossing and $q\to0$ limits eventually required to obtain amplitudes from form factors is actually trivial. In practice, one can therefore perform computations at four points and above directly with amplitudes. A comment along this line was added to the first bullet point on page 4.

A comment was also added in the paragraph between eq. (3) and (4) on page 3, to clarify that the non-analycities in z are directly inherited from those in Mandelstam invariants.

The positions of the non-analyticities at $z=$ or $z\ge M_I^2/s_I$ are independent of the space-time dimension considered.

The fact that all particles in the EFT are massless (combined with dimensional analysis) is heavily used to simplify the dispersive form of $\mathcal{F}{\mathcal{L}}$ after eq. (17). Could you comment on what would happen if small masses would still be present and the integrals would not be scaleless? I would naively expect that the IR branch cut would cancel between the IR and the UV, and the expressions to be unchanged. Is this expectation too naive?

A footnote was introduced on page 5 to explain that massive external states would naively introduce unwanted non-analyticities in z, since one would then need to dilate masses to ensure momentum conservation and on-shell conditions. The case of massless EFTs therefore remains to be examined carefully.

In the derivation of chapter 3 many subtle limits are taken. My understanding is that first the arc is deformed to infinity at $z\to \infty$, then the pole $\delta\to0$ and lastly the limit of $D\to 4$. It would be useful to have a few sentences on this aspect: do these limits commute? The fact that some contours at infinity vanish in dimensional regularization would still be true if first the $D\to 4$ limit was taken?

The $z\to\infty$ limit may introduce UV divergences, while the $\delta\to0$ may introduce IR divergences. In D dimensions, both types can however be regulated simultaneously. Taking the $D\to 4$ limit first would be equivalent to working in four dimensions and other regulators would need to be introduced.

Being able to extract Wilson coefficients from cuts is certainly nice, but an additional phase-space integral (and an integral over z) is still required. From the manuscript is not clear to me if you expect this method to be more computationally powerful than solving directly the loop integration (which has been explored and developed much more). It would be useful to clarify to the reader why this method would be advantageous in practical calculations.

The phase space integrals quickly becomes extremely complicated as the number of uncut loop propagators increases. The cuts of bubble integrals are however trivial. In other cases, the traditional hard-region expansion of the loop is presently evaluated much more easily (for low EFT orders at least) and algorithmically. New methods may in the future help evaluating phase-space integrals, while a combination of the integrand construction from cuts and of hard-region expansion would combine the simplifications arising both approaches. By exhibiting more structure in the matching results, the presented method may still help gaining insight e.g. about possible selection rules. These different aspects are briefly alluded to in our conclusions.

---

## Round 1 · Referee Report · Anonymous · 2023-11-19

Strengths

1. The paper provides a novel solution to the important problem of deriving the (massless) effective field theory
describing the low-energy limit of an ultraviolet theory with massive states. It involves only physical quantities;
objects such as fields, redundancies, etc. are absent.

2. It has the potential of providing not only new insight into matching calculations, but also of making
computations in this respect much more efficient, in the same way as amplitude methods are often much more powerful
than Feynman diagram computations.

3. The paper reproduces previous results about matching, but it also contains certain results to all EFT orders
which, to the best of my knowledge, are new.

4. Despite not being a nuclear part of the work, the paper extends a previous result on the relation between RGE
and the S-matrix, showing that massive particles contribute only to anomalous dimensions above their energy threshold.
(Effectively reproducing the decoupling subtraction scheme.)

5. The paper is generally well written and clear.

Weaknesses

1. Only minor things: some discussions could be clearer, particularly in favor of the non-expert reader.

Report

I think this is a very good work, which clearly deserves publication in SciPost Physics.

Requested changes

I have few minor suggestions that could improve the quality of the work.

1. The authors emphasise the fact that no knowledge on the EFT is needed for the matching. This is definitely a
remarkable difference with respect to usual diagrammatic methods, but isn't it on the same foot as functional methods
in
this respect? If so, I recommend the authors to rather emphasise the main difference with respect to these methods,
which I would say is that no redundant interactions arise (everything is on-shell) and, in fact, all UV
information is encoded in few building blocks (amplitude factorise).
The authors definitely comment on this, but I think it would be clearer if they compare separately with diagrammatic and functional methods.

2. I find the discussion in the big paragraph before equation (2) relatively hard to follow. Is there more than
saying that scaless integrals vanish in dimReg? Providing a very simple example, if possible, might help the
reader follow better this discussion.

3. Equation (9) is the first instance of an all-order result, which is also discussed later (see equation (42)).
Do the authors know
if such results have been derived previously in the literature using other methods? If so, it would be nice to
compare.

4. In the SciPost version of the paper (though not in arXiv), there is a problem with the figure in equation (33).
Please check.

5. Perhaps beyond the scope of the paper (but maybe worth commenting over SciPost?): Could the authors elaborate
a bit on the relation between the methods of regions and the phase-space integrals, as well as on the possibility
of positivity bounds beyond the 2-2 case?

  • validity: high
  • significance: high
  • originality: high
  • clarity: high
  • formatting: excellent
  • grammar: excellent

Author:  Stefano De Angelis  on 2023-12-07  [id 4176]

(in reply to Report 2 on 2023-11-19)

We would like to thank the referee for going carefully through our draft and for the comments. In the following, we would like to address the points he raised.

The authors emphasise the fact that no knowledge on the EFT is needed for the matching. This is definitely a remarkable difference with respect to usual diagrammatic methods, but isn't it on the same foot as functional methods in this respect? If so, I recommend the authors to rather emphasise the main difference with respect to these methods, which I would say is that no redundant interactions arise (everything is on-shell) and, in fact, all UV information is encoded in few building blocks (amplitude factorise). The authors definitely comment on this, but I think it would be clearer if they compare separately with diagrammatic and functional methods.

Our understanding is that the diagrammatic, functional, and on-shell approaches to matching are very similar regarding the information they require about the EFT, at least when the hard-region expansion of loop integrals is employed. In all cases, EFT contact terms (off- or on-shell) are directly obtained from a computation in the UV theory. Then, different techniques can be used to project these results on a specific EFT basis, involving a computation in this basis, or proceeding to a basis reduction of the contact terms obtained. Both paths seem available to us, in all matching approaches. Therefore, we did not mean to contrast our method with either the diagrammatic or functional one. The main feature we highlight, that only non-analyticities are required, is in contrast with both of them.

I find the discussion in the big paragraph before equation (2) relatively hard to follow. Is there more than saying that scaleless integrals vanish in dimReg? Providing a very simple example, if possible, might help the reader follow better this discussion.

The heuristic discussion relative to the vanishing of loop contributions on the EFT side of the power-by-power matching was moved to a footnote to avoid to breaking the flow of this preliminary exposition with details that are not essential at this stage. It now also explicitly refers to the more rigorous discussion that follows.

Equation (9) is the first instance of an all-order result, which is also discussed later (see equation (42)). Do the authors know if such results have been derived previously in the literature using other methods? If so, it would be nice to compare.

We do not know whether such a result was previously presented in the literature. An all-order comparison is possible with the expansion of the exact loop integral evaluated in the hard region using the geometrical method of regions (instead of the more pedestrian order-by-order expansion of the loop integrand in powers of the external momenta).

In the SciPost version of the paper (though not in arXiv), there is a problem with the figure in equation (33). Please check.

We will verify that our resubmission displays this figure properly.

Perhaps beyond the scope of the paper (but maybe worth commenting over SciPost?): Could the authors elaborate a bit on the relation between the methods of regions and the phase-space integrals, as well as on the possibility of positivity bounds beyond the 2-2 case?

Unfortunately, we currently have no concrete statement to make about either of these possible direction of future explorations.

---

## Round 2 · Referee Report · Anonymous (Referee 2) · 2023-12-13

Report

The authors have addressed the points I made in the report. I recommend the paper for publication in its current form.

---

## Round 2 · Author Response

We simply address the point raised by the referees.

---

## Round 2 · List of Changes

• In pag. 3 we added a sentence between eq. (3) and eq. (4), and footnote 1 has been moved from the main text.
  • In pag. 4 we added a sentence to the first bullet point.
  • In pag. 5 we added footnote 2.
  • In pag. 8, eq. (22) has been changed and two related formulae in the text following eq. (23).

---

## Editorial Decision

published